# Genetic Insights into Intestinal Microbiota and Risk of Infertility: A Mendelian Randomization Study

**DOI:** 10.3390/microorganisms11092319

**Published:** 2023-09-15

**Authors:** Fuxun Zhang, Yang Xiong, Kan Wu, Linmeng Wang, Yunhua Ji, Bo Zhang

**Affiliations:** 1Department of Urology, Tangdu Hospital, Air Force Medical University, Xi’an 710038, China; 2Department of Urology, Institute of Urology, West China Hospital, Sichuan University, Chengdu 610041, China

**Keywords:** intestinal microbiota, infertility, causal association, mendelian randomization, mechanism

## Abstract

Background: The interaction between intestinal microbiota and infertility is less researched. This study was performed to investigate the causal association between gut microbiota and infertility. Methods: In this two-sample Mendelian randomization (MR) study, genetic variants of intestinal microbiota were obtained from the MiBioGen consortium, which included 18,340 individuals. Inverse variance weighting (IVW), MR-Egger, weighted median, maximum likelihood, MR Robust adjusted profile score, MR Pleiotropy residual sum, and outlier (MR-PRESSO) methods were used to explore the causal links between intestinal microbiota and infertility. The MR-Egger intercept term and the global test from the MR-PRESSO estimator were used to assess the horizontal pleiotropy. The Cochran Q test was applied to evaluate the heterogeneity of instrumental variables (IVs). Results: As indicated by the IVW estimator, significantly protective effects of the *Family XIII AD3011 group* (OR = 0.87) and *Ruminococcaceae NK4A214 group* (OR = 0.85) were identified for female fertility, while *Betaproteobacteria* (OR = 1.18), *Burkholderiales* (OR = 1.18), *Candidatus Soleaferrea* (OR = 1.12), and *Lentisphaerae* (OR = 1.11) showed adverse effects on female fertility. Meanwhile, *Bacteroidaceae* (OR = 0.57), *Bacteroides* (OR = 0.57), and *Ruminococcaceae NK4A214 group* (OR = 0.61) revealed protective effects on male fertility, and a causal association between *Anaerotruncus* (OR = 1.81) and male infertility was detected. The effect sizes and directions remained consistent in the other five methods except for *Candidatus Soleaferrea*. No heterogeneity or pleiotropy were identified by Cochran’s Q test, MR-Egger, and global test (all *p* > 0.05). Conclusions: This two-sample MR study revealed that genetically proxied intestinal microbiota had potentially causal effects on infertility. In all, the *Ruminococcaceae NK4A214 group* displayed protective effects against both male and female infertility. Further investigations are needed to establish the biological mechanisms linking gut microbiota and infertility.

## 1. Background

The intestinal microbiota has been regarded as the second human genome, mediating the development of various conditions [1]. The intestinal microbiome has many important roles in digestion, drug metabolism, immunity training, endocrine regulation, and so on, which may affect the host’s health through different pathways [2]. Currently, abnormal composition of intestinal microbiota has become a significant feature shared by many different pathogenesis [3]. It is suggested that gut microbiota might be a risk or preventive factor for many diseases, including metabolic conditions and cancer [3,4]. Meanwhile, gut microbiota is reportedly associated with multiple gynecologic conditions and female infertility [5]. Moreover, the metabolites of the gut microbiota may affect testicular function and male fertility [6]. However, the role of intestinal microbiota in infertility is less studied.

Infertility is a prevalent and intricate reproductive disorder that affects approximately 8–12% of couples worldwide [7]. This disease can have profound emotional, physical, and social impacts on couples, leading to a heavy psychological burden [8]. Most infertile couples have identifiable causes, such as ovulatory dysfunction, male factor infertility, and tubal disease; however, approximately 15% of infertile couples have unexplained causes [9]. Thus, understanding the causes of infertility and exploring the causal link between risk factors and infertility are crucial for affected couples and healthcare professionals. In all, it has been documented that microbial dysbiosis of the genital tract incurred by various factors represents a risk factor for infertility [10]. However, the causal association of gut microbiota with infertility remains unclear.

Theoretically, a randomized controlled trial (RCT) is the best method to establish a causal relationship between exposure and disease. However, it is not realistic to explore the causal association of intestinal microbiota with infertility using an RCT because of too many objective limitations and confounding factors. As a novel epidemiological method, Mendelian randomization (MR) uses single nucleotide polymorphisms (SNPs) as instrumental variables (IVs) to replace exposures and outcomes. This method has been widely used for causal inferences, avoiding confounding bias and reverse causality from a limited sample size and cross-sectional design [11,12]. Thus, we performed this two-sample MR analysis to explore the causal link between intestinal microbiota and infertility utilizing the genome-wide association study (GWAS) summary statistics.

## 2. Methods

### 2.1. Data Sources and Selection of Instrumental Variable

The flowchart of this study is shown in Figure 1. Summary statistics on human intestinal microbiota were obtained from the MiBioGen consortium [13]. As a retrospective design, this study included 18,340 individuals from 24 cohorts. The microbial composition and relative abundances of the intestinal microbiota were detected by using 16S ribosomal RNA gene sequencing. A microbiota quantitative trait loci mapping analysis was conducted to identify genetic variants in the host and evaluate the association between human genetic variants and intestinal microbiota. In this study, the lowest taxonomic level was genus. Furthermore, GWAS summary statistics of infertility (male and female) were retrieved from FinnGen (https://r9.finngen.fi/). The summary statistics for male infertility included 1271 cases and 119,297 controls, and those for female infertility included 13,142 cases and 107,564 controls (Appendix A). The diagnosis of infertility was based on the International Classification of Diseases (ICD) codes. The diagnostic codes were 606 (ICD-8 and ICD-9) and N46 (ICD-10) for male infertility and N97 (ICD-10) for female infertility (Appendix A). All the participants were of European descent.

### 2.2. Selection of Instrumental Variables

To obtain adequate IVs and increase the statistical power, IVs were filtered from the identified SNPs at a genome-wide statistical significance of *p* < 1 × 10^−5^, as previous studies did [14]. The left SNPs were further pruned if the linkage disequilibrium *r*^2^ was ≥0.01 at a window size of 10,000 kb. SNPs with a minor allele frequency (MAF) < 0.01 are generally accepted as rare SNPs, which have limited impact on the traits. Therefore, only SNPs with MAF ≥ 0.01 were reserved.

### 2.3. Statistical Analysis

Six methods were used to explore the causal effects of intestinal microbiota on infertility, including inverse variance weighted (IVW), MR-Egger, weighted median, maximum likelihood (ML), MR robust adjusted profile score (MR.RAPS), and MR pleiotropy residual sum and outlier (MR-PRESSO). In IVW models, all IVs are assumed to be valid and then combined using a meta-technique. This method is used as the main analysis due to its high statistical power. When heterogeneity existed, the random effects IVW model was used; otherwise, the fixed effects IVW estimator was adopted as the main analysis. To generate unbiased results even when pleiotropy existed, MR-Egger and weighted median methods relaxed this assumption and were used as sensitivity analyses. In addition, the ML method was also used for its minimal bias in limited sample sizes. The MR.RAPS method could produce consistent results when weak and pleiotropic SNPs exist. After excluding pleiotropic outliers, the MR-PRESSO estimator combined the effects from IVs into the IVW model. This estimator could be used to detect the presence of pleiotropy. The recalculated genetic association was used as the outcome data in the sensitivity analysis.

The Cochran Q test was applied to evaluate the heterogeneity of IVs. Q statistics with a *p*-value < 0.05 indicated the presence of heterogeneity, and the random-effects IVW method was used to generate more conservative but robust estimates. To assess the horizontal pleiotropy, the MR-Egger intercept term and the global test from the MR-PRESSO estimator were used. The strength of SNPs was quantified by calculating the F-statistics of each bacterial taxon, as previously reported [15]. A F-statistic greater than 10 indicated a lower likelihood of weak instrumental bias.

All statistical analyses were conducted using R 4.0.3 (R Foundation for Statistical Computing, Vienna, Austria). “TwoSampleMR”, “mr.raps”, and “MR-PRESSO” packages were used for data analyses. The “forest” package was used to draw the forest plot.

## 3. Results

### 3.1. Protective Effects of Genetically Proxied Microbiota on Female Fertility

Two germs showed protective effects on female fertility (Figure 2). A total of 13 SNPs were used as IVs in the *Family XIII AD3011 group*, with an average F statistic of 21.47 (Table 1). A standard deviation (SD) increment in the abundance of the genetically proxied *Family XIII AD3011 group* led to a reduced risk of female infertility (OR: 0.87, 95% CI: 0.77–0.99, *p* for IVW = 0.041) (Figure 2). The results remained significant when ML and MR-PRESSO were used (all *p* < 0.05). Meanwhile, the direction and effect size of other estimators were similar to IVW (Figure 2). The scatter plot displayed a decreasing risk of female infertility with the increase of the SNP effect on the *Family XIII AD3011 group* (Figure 3D).

The *Ruminococcaceae NK4A214 group* is a genus of *Ruminococcaceae*, and a total of 13 SNPs were used as IVs of this germ with an average F statistic of 21.66 (Table 1). The IVW estimator detected a protective effect of the *Ruminococcaceae NK4A214 group* against female infertility (OR: 0.85, 95% CI: 0.74–0.98, *p* for IVW = 0.021) (Figure 2). This result remained significant in ML, RAPS, and MR-PRESSO (all *p* < 0.05). The scatter plots showed decreased risks of female infertility with the increase of the SNP effect on the *Ruminococcaceae NK4A214 group* (Figure 3F). Moreover, no pleiotropy or heterogeneity were detected for these two germs (all *p* > 0.05) (Figure 4, Table 1 and Table 2).

### 3.2. Adverse Effects of Genetically Proxied Microbiota on Female Fertility

Four germs showed adverse effects on female fertility (Figure 2). There were 10 SNPs used as IVs in *Betaproteobacteria* with an average F statistic of 21.78 (Table 1). The IVW estimator revealed an adverse effect of *Betaproteobacteria* on female fertility (OR: 1.18, 95% CI: 1.01–1.38, *p* for IVW = 0.037) (Figure 2). The results were still significant in ML, RAPS, and MR-PRESSO (all *p* < 0.05), and the direction of other estimators was consistent with IVW. With the increase in SNP effects on *Betaproteobacteria*, the scatter plot showed an increasing risk of female infertility (Figure 3A).

At the order level, a total of 10 SNPs were used as IVs of *Burkholderiales*, with an average F statistic of 22.27 (Table 1). The IVW estimator showed that increased abundance of genetically proxied *Burkholderiales* led to an increased risk of female infertility (OR: 1.18, 95% CI: 1.01–1.38, *p* for IVW = 0.034) (Figure 2). The results remained significant when ML, RAPS, and MR-PRESSO were used (all *p* < 0.05), and the findings of other estimators did not reveal discordance. Genetically proxied *Burkholderiales* abundance was positively related to the risk of female infertility (Figure 3B). No heterogeneity or pleiotropy in MR analyses were observed (all *p* > 0.05) (Figure 4, Table 1 and Table 2). A total of 9 SNPs were used as IVs in *Candidatus Soleaferrea*, with an average F statistic of 20.78 (Table 1). The IVW estimator revealed a potentially causal association of *Candidatus Soleaferrea* with female infertility (OR: 1.12, 95% CI: 1.01–1.25, *p* for IVW = 0.048) (Figure 2). This finding remained consistent in ML, RAPS, and MR-PRESSO. With the increase in SNP effect on *Candidatus Soleaferrea*, the effect of SNP on infertility ascended (Figure 3C).

At phylum level, there were 9 SNPs used as IVs in *Lentisphaerae* with an average F statistic of 22.05 (Table 1). At phylum level, the causal relationship between *Lentisphaerae* and female infertility was identified (OR: 1.11, 95% CI: 1.01–1.20, *p* for IVW = 0.044) (Figure 2). The directions of other estimators were consistent with IVW. The scatter plots displayed increased risks of female infertility with the increase in SNP effects on *Lentisphaerae* (Figure 3E). Additionally, no pleiotropy or heterogeneity were detected for these germs (all *p* > 0.05) (Figure 4, Table 1 and Table 2).

### 3.3. Protective Effects of Genetically Proxied Microbiota on Male Fertility

Three germs showed protective effects on male fertility. At family level, there were 8 SNPs used as IVs in *Bacteroidaceae* with an average F statistic of 22.29 (Table 1). The IVW estimator revealed a protectively causal effect of *Bacteroidaceae* against male infertility (OR: 0.57, 95% CI: 0.33–0.96, *p* for IVW = 0.036) (Figure 5). This finding was supported by ML, RAPS, and MR-PRESSO (all *p* < 0.05). The scatter plots showed that with the increment of SNP effects on the abundances of *Bacteroidaceae*, SNP effects on male infertility decreased (Figure 6B). No pleiotropy was identified by the global test and MR-Egger method (all *p* > 0.05) (Table 1), and no heterogeneity was detected by the Cochran Q test according to the funnel plot (all *p* > 0.05) (Figure 7, Table 2).

At genus level, a total of 8 SNPs were used as IVs in *Bacteroides* with an average F statistic of 22.29 (Table 1). The protective effect of genetically proxied *Bacteroides* on male fertility was identified by the IVW estimator (OR: 0.57, 95% CI: 0.33–0.96, *p* for IVW = 0.036) (Figure 5). This causality was further supported by ML, RAPS, and MR-PRESSO (all *p* < 0.05). Scatter plots visualizing the increased SNP effect on *Bacteroides* and the decreased SNP effect on male infertility were displayed (Figure 6C). No pleiotropy or heterogeneity was detected by the global test, the MR-Egger method, or the Cochran Q test (all *p* > 0.05) (Figure 7, Table 1 and Table 2).

There were 13 SNPs used as IVs in the *Ruminococcaceae NK4A214 group* with an average F statistic of 21.66 (Table 1). The IVW estimator disclosed the protective effect of the *Ruminococcaceae NK4A214 group* on male fertility (OR: 0.61, 95% CI: 0.39–0.97, *p* for IVW = 0.037). This finding was consistent with ML, RAPS, and MR-PRESSO (all *p* < 0.05). Of note, the protective role of the *Ruminococcaceae NK4A214 group* was also detected in female infertility, suggesting that *Ruminococcaceae NK4A214* may be a core genus affecting fertility. All directions and effect sizes of the MR-Egger and weighted median for the three above-mentioned germs were in accordance with the IVW estimator. The scatter plots showed decreased risks of male infertility with the increase of the SNP effect on the *Ruminococcaceae NK4A214 group* (Figure 6D). No pleiotropy was detected by the global test and MR-Egger method (all *p* > 0.05) (Table 1), and no heterogeneity was identified by the Cochran Q test according to the funnel plot (all *p* > 0.05) (Figure 7, Table 2).

### 3.4. Adverse Effects of Genetically Proxied Microbiota on Male Fertility

There was only one germ that had an adverse causal effect on male fertility. A total of 13 SNPs were used as IVs of *Anaerotruncus*, with an average F statistic of 20.84 (Table 1). With an increase in SD in the abundance of genetically proxied *Anaerotruncus*, an increased risk of male infertility was detected (OR: 1.81, 95% CI: 1.14–0.99, *p* for IVW = 0.011) (Figure 5). This causality was also detected by ML, RAPS, and MR-PRESSO (all *p* < 0.05). The directions of other estimators were consistent with IVW. The scatter plots displayed increased risks of male infertility with the increase in SNP effects on *Anaerotruncus* (Figure 6A). No pleiotropy was identified by the global test and MR-Egger method, and the Cochran Q test detected no heterogeneity according to the funnel plot (*p* > 0.05) (Figure 7, Table 1 and Table 2).

## 4. Discussion

Infertility is a major health issue in the world, which will lead to a severe decline in child births and considerable population declines [16]. Meanwhile, infertility is a multi-factorial pathological condition with high complex and heterogeneous etiologies [17]. Although many risk factors have been documented, the interaction between gut microbiota and infertility requires further investigation. As the second genome of humans, the gut microbiota plays an important role in the pathogenesis of many diseases [18]. The gut microbiota may participate in the development of many conditions by producing microbial metabolites and subsequently activating downstream signaling pathways, which eventually change cellular function and initiate the abnormality [19]. Currently, the relationship between gut microbiota and infertility remains unclear. Here, we used genetic data to explore the causal link between intestinal microbiota and infertility in order to avoid bias from confounding factors.

In our MR analysis, protective effects of the *Family XIII AD3011 group* and the *Ruminococcaceae NK4A214 group* on female fertility were identified, while *Betaproteobacteria*, *Burkholderiales*, *Candidatus Soleaferrea*, and *Lentisphaerae* showed adverse effects on female fertility. At the same time, *Bacteroidaceae*, *Bacteroides*, and *Ruminococcaceae NK4A214 group* revealed protective effects on male fertility, while an adverse causal association between *Anaerotruncus* and male fertility was detected. This study identified causal links between intestinal microbiota and female or male infertility, offering novel insights into the assessment of potential causes of infertility.

At the genus level, previous evidence supporting the protective role of the *Family XIII AD3011 group* in female fertility is scarce. In a case-control study, the abundance of the *Family XIII AD3011 group* was found to be related to several markers of polycystic ovary syndrome (PCOS) [20]. Meanwhile, it is reported that the *Family XIII AD3011 group* was negatively correlated with adipic acid, affecting the metabolic phenotype and host inflammation [21]. Thus, the protectively causal effect of the *Family XIII AD3011 group* on female fertility may be involved in the inhibition of PCOS-related molecular expression and inflammatory signaling, which may impair the function of reproductive organs.

The *Ruminococcaceae NK4A214 group* is a genus of *Ruminococcaceae*. It should be noted that the *Ruminococcaceae NK4A214 group* in our study showed protective influences on both female and male fertility, suggesting a potentially core role for the *Ruminococcaceae NK4A214 group* in the pathogenesis of infertility. Reduced abundance of the *Ruminococcaceae NK4A214 group* may result in abnormal spermatogenesis through reduced bile acid levels and vitamin A absorption [22]. For female infertility, significantly decreased *Ruminococcaceae* were seen in the PCOS group compared with healthy controls [23]. Meanwhile, increasing commensal *Ruminococcaceae* via fecal microbiota transplantation (FMT) may improve the ovarian function of aged mice by inhibiting pro-inflammatory interferon (TNF)-γ signaling and promoting anti-inflammatory interleukin (IL)-4 signaling [24]. Taken together, the *Ruminococcaceae NK4A214 group* might have a crucially protective effect against infertility through increasing vitamin A absorption, inhibiting inflammatory signaling, and PCOS-related genetic expression.

*Betaproteobacteria* are a class of *Proteobacteria*. At the phyla level, a higher abundance of *Proteobacteria* in peritoneal fluid could be seen among infertile patients with endometriosis, in which several inflammatory factors, including IL-6, IL-10, IL-13, and TNF-α, may be involved [25]. Moreover, it is suggested that *Betaproteobacteria* and *Proteobacteria* might mediate the effects of estrogen on peripheral organs [26]. Thus, *Betaproteobacteria* may adversely affect female fertility through inflammatory and estrogen signaling pathways, impairing the function of reproductive organs and interfering with processes such as ovulation, implantation, and embryonic development. *Burkholderiales* are a class of bacteria known to have both beneficial and pathogenic effects on human health [27]. A molecular analysis showed that *Burkholderiales* was detectable in the endometrial fluid of infertile women and was negative in fertile controls [28]. Meanwhile, some genera of *Burkholderiales* might be associated with cystic fibrosis and could cause a broad range of infections in hosts [29]. Thus, we hypothesize that inflammatory signaling and fibrotic mechanisms might underlie the causal association of *Burkholderiales* with female infertility.

At the genus level, *Candidatus Soleaferrea* was observed to be related to several autoimmune diseases via the effects of metabolites and the immune-gut axis [30]. However, the evidence related to the link between *Candidatus Soleaferrea* and female infertility is scarce. Whether the causal effect of *Candidatus Soleaferrea* on infertility depends on the immune-gut axis needs further investigation. It is reported that patients with lung cancer had higher levels of *Lentisphaerae* than the healthy controls, and certain specific bacteria were correlated with serum inflammatory indicators in those patients, indicating the potential links between *Lentisphaerae* and systemic immunity and inflammation in hosts [31]. Meanwhile, a decreased abundance of *Lentisphaerae* was detected in patients with autoimmune hepatitis compared to healthy controls [32]. These results suggested that *Lentisphaerae* and its metabolites might be associated with abnormal immune responses or suppression, which may potentially impact fertility-related processes.

*Bacteroides* is a type genus of the family *Bacteroidaceae*. In our MR analysis, both *Bacteroides* and *Bacteroidaceae* showed protective effects on male fertility, indicating the potential role of *Bacteroides* in the management of infertile males. It has been found that *Bacteroides* might affect the iron metabolism in infertile females, highlighting the links between *Bacteroides* and reproductive function [33]. Several species of *Bacteroides* were detected to be correlated with non-obstructive azoospermia in male infertility [34]. Meanwhile, fecal microbiota transplantation may improve the semen quality of infertile males with a high-fat diet by increasing gut *Bacteroides*, improving liver function, and ameliorating the testicular microenvironment [35]. All in all, the influence of *Bacteroides* on male fertility might be positive, in which the gut microbiota-testis axis and spermatogenesis signaling may be involved.

*Anaerotruncus* is a genus assigned to the family *Ruminococcaceae*. As a conditional pathogenic bacterium, *Anaerotruncus* has an important role in the maintenance of microbial diversity, which is crucial to host energy homeostasis affected by a high-fat/sucrose diet [36]. Meanwhile, *Anaerotruncus* was reportedly associated with the formation of hepatocellular carcinoma related to non-alcoholic fatty liver disease [37]. Similarly, decreased abundance of *Anaerotruncus* was considered to mediate the therapeutic effect of liraglutide on fatty liver in db/db mice [38]. Thus, the adverse effect of *Anaerotruncus* on male fertility may be due to disrupted energy homeostasis and glucolipid metabolism, which might interfere with male fertility-related processes.

Several limitations in this study should be noted when explaining the results. Firstly, the trans-ethnic population (mainly Europeans) was included in the dataset to obtain the genetic links between intestinal microbiota and SNPs. The mixed ethnic background may produce bias to the genetic association. Further identification with a simpler ethnic background and a larger sample size is necessary. Secondly, the abundance of gut microbiota is easily affected by many environmental factors, such as diet and drugs. Meanwhile, the dataset originates from 24 cohorts in different countries. Therefore, different environmental factors among different cohorts may produce significant heterogeneity and bias the genetic association. Thirdly, due to the limitations of summary-level statistics, the combined role of several risky germs cannot be determined, which may need further investigation. Additionally, the etiology and risk factors of infertility are complex and often mixed, and most links between gut microbiota and different causes remain unknown. In this study, the causes of infertility were diagnosed by ICD codes as in other Mendelian randomization studies, but the results were not adjusted for documented causes or etiologies, such as hormonal profiles, karyotype, and spemogram data, which may affect the conclusion and should be investigated in the future.

## 5. Conclusions

In this study, we found that the genetically proxied *Family XIII AD3011 group*, *Ruminococcaceae NK4A214 group*, *Betaproteobacteria*, *Burkholderiales*, *Candidatus Soleaferrea*, and *Lentisphaerae* had potentially causal effects on female infertility. Meanwhile, genetically proxied *Bacteroidaceae*, *Bacteroides*, *Ruminococcaceae NK4A214 group*, and *Anaerotruncus* had potentially causal effects on male infertility. Among them, the *Ruminococcaceae NK4A214 group* was found to be the core germ, which has a protective effect on both female and male infertility. Further investigations are needed to identify the biological mechanisms linking gut microbiota and infertility.

## Figures and Tables

**Figure 1 microorganisms-11-02319-f001:**
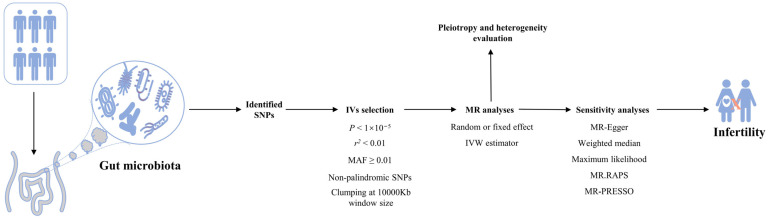
Schematic diagram. Abbreviations: IVs—instrumental variables; IVW—inverse variance weighted; MR—Mendelian randomization; MAF—minor allele frequency; PRESSO—pleiotropy residual sum and outlier; RAPS—robust adjusted profile score; SNPs—single nucleotide polymorphisms.

**Figure 2 microorganisms-11-02319-f002:**
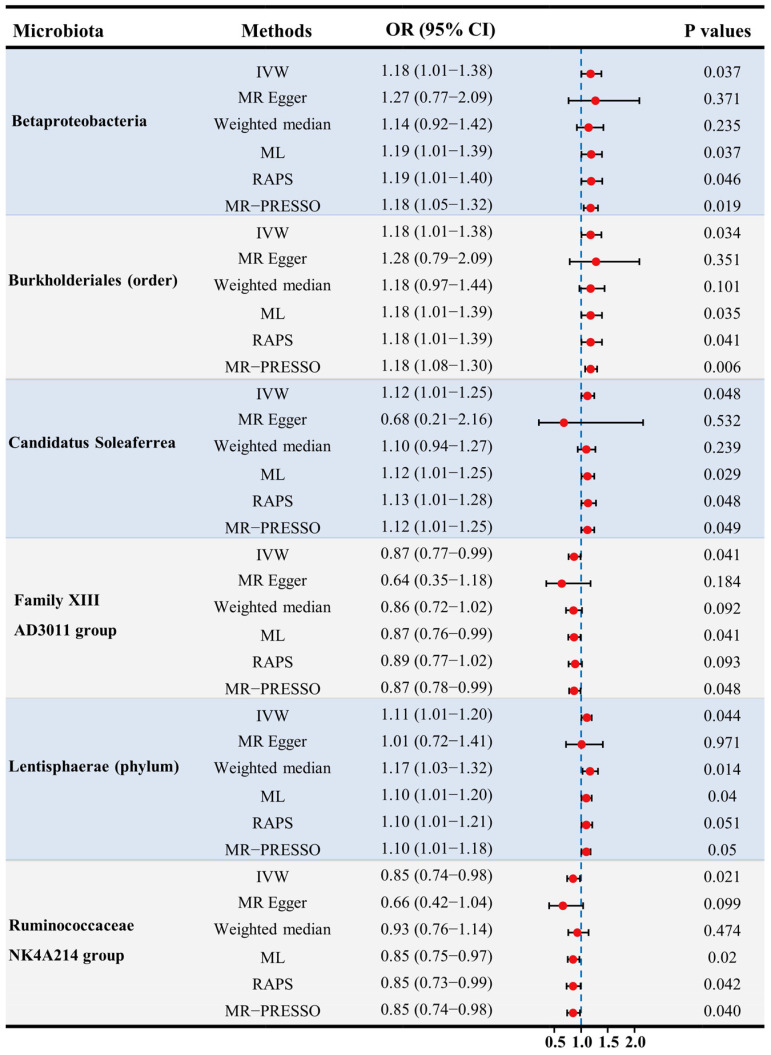
The results of MR estimating the causal association between intestinal microbiota and female infertility. Abbreviations: CI—confidence interval; IVW—inverse variance weighted; MR—Mendelian randomization; OR—odds ratio; PRESSO—pleiotropy residual sum and outlier; RAPS—robust adjusted profile score.

**Figure 3 microorganisms-11-02319-f003:**
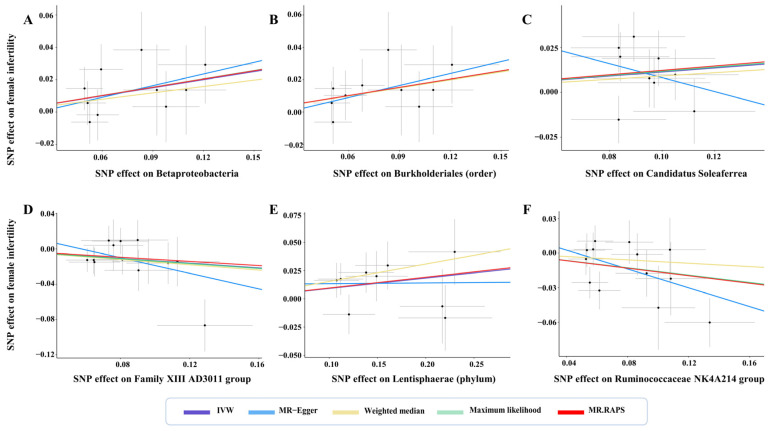
Scatter plots for the causal effects of gut microbiota on female infertility. Abbreviations: IVW—inverse variance weighted; RAPS—robust adjusted profile score.

**Figure 4 microorganisms-11-02319-f004:**
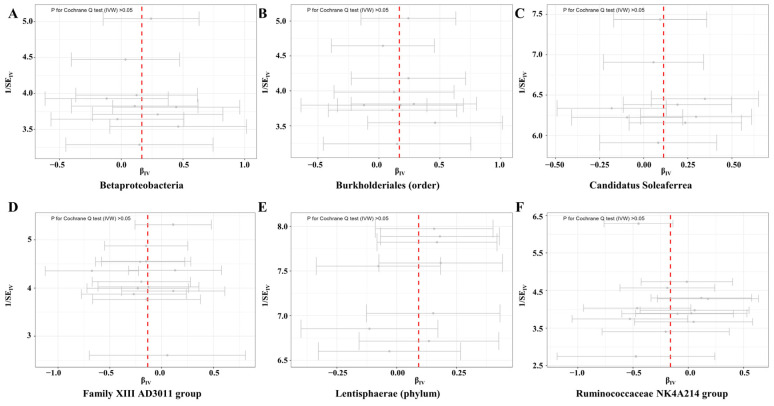
Funnel plots for the heterogeneity of instrumental variables for female infertility. Abbreviations: IVW—inverse variance weighted; SE—size effect.

**Figure 5 microorganisms-11-02319-f005:**
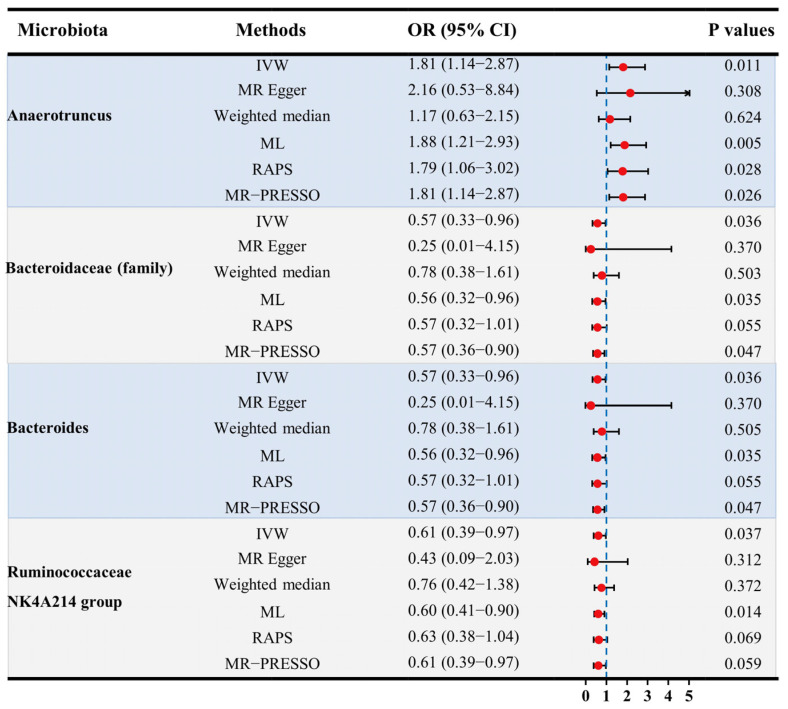
The results of MR estimating the causal association between intestinal microbiota and male infertility. Abbreviations: CI—confidence interval; IVW—inverse variance weighted; MR—Mendelian randomization; OR—odds ratio; PRESSO—pleiotropy residual sum and outlier; RAPS—robust adjusted profile score.

**Figure 6 microorganisms-11-02319-f006:**
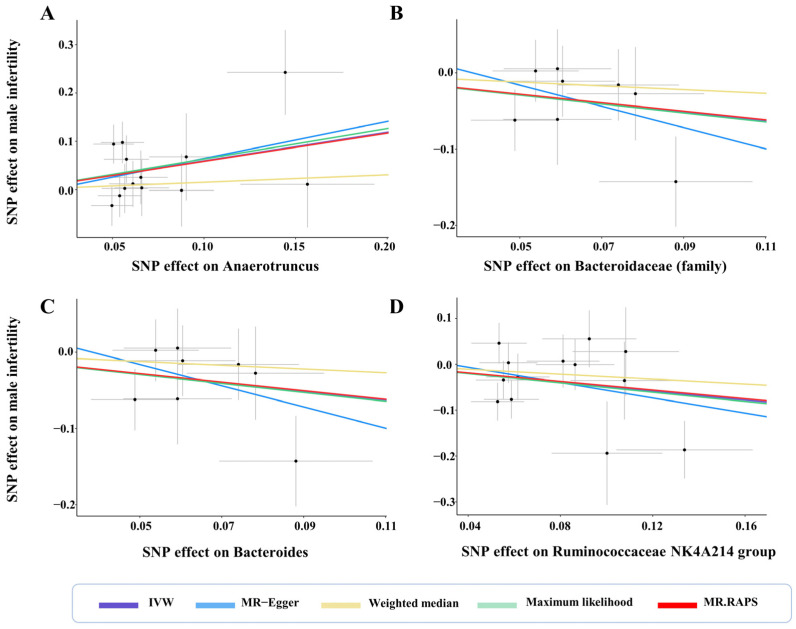
Scatter plots for the causal effects of gut microbiota on male infertility. Abbreviations: IVW—inverse variance weighted; RAPS—robust adjusted profile score.

**Figure 7 microorganisms-11-02319-f007:**
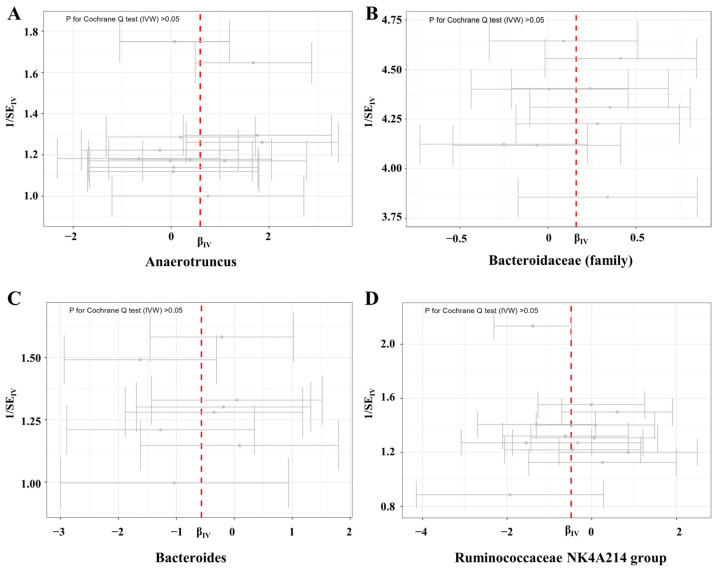
Funnel plots for the heterogeneity of instrumental variables for female infertility. Abbreviations: IVW—inverse variance weighted; SE—size effect.

**Table 1 microorganisms-11-02319-t001:** Heterogeneity and pleiotropy of instrumental variables.

Traits	Family/Genus	Number of SNPs	F-Statistics	*p* Values (Global Test)	Intercept (Egger Test)	*p* Values (Egger Test)
Male infertility	*Anaerotruncus*	13	20.84	0.302	−0.0126	0.801
	*Bacteroidaceae (family)*	8	22.29	0.644	0.0534	0.580
	*Bacteroides*	8	22.29	0.625	0.0534	0.580
	*Ruminococcaceae NK4A214 group*	13	21.66	0.166	0.0268	0.652
Female infertility	*Betaproteobacteria*	10	21.78	0.850	−0.0054	0.767
	*Burkholderiales (order)*	10	22.27	0.963	−0.0059	0.744
	*Candidatus Soleaferrea*	9	20.78	0.320	0.0470	0.422
	*Family XIII AD3011 group*	13	21.47	0.621	0.0254	0.335
	*Lentisphaerae (phylum)*	9	22.05	0.655	0.0127	0.626
	*Ruminococcaceae NK4A214 group*	13	21.66	0.330	0.0190	0.277

Abbreviations: SNPs—single nucleotide polymorphisms.

**Table 2 microorganisms-11-02319-t002:** Results of Cochrane’s Q tests.

Traits	Family/Genus	Q Values			*p* Values		
		MR Egger	IVW	ML	MR Egger	IVW	ML
Male infertility	*Anaerotruncus*	13.982	14.068	13.633	0.234	0.296	0.325
	*Bacteroidaceae (family)*	4.931	5.273	5.129	0.553	0.627	0.644
	*Bacteroides*	4.931	5.273	5.129	0.553	0.627	0.644
	*Ruminococcaceae NK4A214 group*	15.958	16.270	15.901	0.143	0.179	0.196
Female infertility	*Betaproteobacteria*	4.690	4.785	4.696	0.790	0.853	0.860
	*Burkholderiales (order)*	3.091	3.205	3.146	0.928	0.956	0.958
	*Candidatus Soleaferrea*	8.852	9.771	9.533	0.263	0.281	0.299
	*Family XIII AD3011 group*	9.325	10.343	10.175	0.592	0.586	0.601
	*Lentisphaerae (phylum)*	5.863	6.122	6.016	0.556	0.634	0.645
	*Ruminococcaceae NK4A214 group*	12.021	13.449	13.193	0.362	0.337	0.355

Abbreviations: MR—Mendelian randomization; IVW—inverse variance weighting; ML—maximum likelihood.

## Data Availability

The datasets analyzed during the current study are available from the corresponding authors upon reasonable request.

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
