# Peer review of "Genetic Insights into Intestinal Microbiota and Risk of Infertility: A Mendelian Randomization Study"

_microorganisms, 2023, doi:10.3390/microorganisms11092319_

Round 1
Reviewer 1 Report
Topics related to the intestinal microbiota are becoming increasingly the subject of scientific study.
The paper presented to me for review, "Genetic insights into intestinal microbiota and risk of infertility: a Mendelian randomization study," touches on the medically interesting issue of gut microbiota and infertility.
The work was planned correctly.
The study used the right research methods for the stated goals.
In their work, the authors used appropriate advanced statistical methods
The work contains 6 figures and 2 tables.
It should be noted that the authors drew interesting and adequate conclusions on the basis of their research.
The authors cited 38 items of current literature.
In my opinion, the work can be published in its present form.
Author Response
Thank you for reviewing my manuscript, and providing us so many valuable and kind comments. We all believe that your valuable comments will encourage us to conduct subsequent research and exploration in this field afterwards. Meanwhile, we carefully checked the paper again and made some necessary changes to the grammar and expression. Please see the revisions noted as yellow color in the manuscript. We hope our work will meet your requirements and let you feel satisfied.
Reviewer 2 Report
In this Manuscript the authors use gene sequencing data to try to determine the relevance of microbiome towards both male and female human infertility. While this is potentially interesting there are several issues that need to be fully addressed before the data itself can even begin to be analyzed:
1- I assume this is a retrospective study, not a prospective one (i.e. the authors used available databases but performed no sequencing themselves), although this is not specifically stated in the text, as it should be.
2- The authors state that male infertility included 1,271 cases and 119,297 controls, and that for female infertility there were 13,142 cases and 107,564 controls. Were all these cases compared? It seems that, not only is this a huge amount of data, but that there is a great imbalance cases-controls that could warp the data analysis. How were the cases identified and selected? More details and rationale should be provided at this level.
3- Most importantly: saying that patients suffer from “infertility” is the same as saying they suffer from “cancer”, for example. There are many types of infertility (both male and female) and they are not necessarily similar. Therefore, when comparing cases to controls it is extremely likely (indeed almost mandatory) that the authors are lumping together very distinct causes of infertility. This must absolutely be addressed. What kinds of infertility are the authors discussing? What were the inclusion/exclusion criteria for them? What characteristics do the patients have (age, habits, spemogram and eco data, hormonal profiles, Karyotype, etc etc etc) “Infertility” alone is not a good criterion. A thorough table including all the relevant information on the patients used must be included, other wise it is completely impossible to even begin to consider the potential relevance of the data.
Round 2
Reviewer 2 Report
The authors have adequately addressed my concerns, I have no further comments.